# Hepatitis B Co-Infection Has Limited Impact on Liver Stiffness Regression in Chronic Hepatitis C Patients Treated with Direct-Acting Antivirals

**DOI:** 10.3390/v14040786

**Published:** 2022-04-10

**Authors:** Cheng-Er Hsu, Yen-Chun Liu, Ya-Ting Cheng, Wen-Juei Jeng, Rong-Nan Chien, Chun-Yen Lin, Dar-In Tai, I-Shyan Sheen

**Affiliations:** 1Department of Gastroenterology and Hepatology, Chang Gung Memorial Hospital Linkou Medical Center, Taoyuan 333, Taiwan; sjuujs2001@cgmh.org.tw (C.-E.H.); b9602082@cgmh.org.tw (Y.-C.L.); missbear123@cgmh.org.tw (Y.-T.C.); ronald@cgmh.org.tw (R.-N.C.); chunyenlin@cgmh.org.tw (C.-Y.L.); tai48978@cgmh.org.tw (D.-I.T.); happy95kevin@gmail.com (I.-S.S.); 2College of Medicine, Chang Gung University, Taoyuan 333, Taiwan

**Keywords:** HBV reactivation, hepatocellular carcinoma, hepatic decompensation

## Abstract

Introduction: High sustained virological response (SVR) rate (>95%) and liver stiffness regression can be achieved with direct acting antivirals treatment (DAA) in patients with chronic hepatitis C virus (CHC) infection. Reactivation of hepatitis B virus (HBV) was reported during DAA treatment in patients co-infected with HBV, although its impact on liver stiffness remains unknown. This study aims to investigate whether the liver stiffness (LSM) regression is different between HBV/HCV co-infected and mono-HCV-infected patients. Materials and Methods: CHC patients with/without HBV co-infection who received DAA treatment and achieved SVR12 between March 2015 and December 2019 in Chang Gung Memorial Hospital, Linkou branch were prospectively enrolled. LSM was assessed by transient elastography (TE, Fibroscan) at baseline and after SVR. Propensity score matching (PSM) at 3:1 ratio, adjusted for age, gender, pre-DAA alanine aminotransferase (ALT), platelet count, and LSM, between CHC with and without HBV co-infection, was performed before further analysis. Results: Among 906 CHC patients enrolled, 52 (5.7%) patients had HBV/HCV co-infection. Patients with HBV/HCV co-infection were of younger age (61.8 vs. 63.2, *p* = 0.31), with a higher proportion of males (53.8% vs. 38.9%, *p* = 0.03), and lower pretreatment LSM level (8.15 vs. 10.2 kPa, *p* = 0.09), while other features were comparable. After PSM, patients with HBV/HCV co-infection had insignificantly lower LSM regression compared to mono-HCV-infected patients (−0.85 kPa vs. −1.65 kPa, *p* = 0.250). Conclusions: The co-infection of HBV among CHC patients has limited impact on liver stiffness regression after successful DAA treatment.

## 1. Introduction

Hepatitis B virus (HBV) and hepatitis C virus (HCV) are the two major leading causes of liver cirrhosis and hepatic cellular carcinoma (HCC) [1,2], resulting in serious health problems worldwide. In recent years, direct-acting antivirals (DAA) have shown the potential for HCV eradication through high sustained virological response (SVR) rates (>95%) even in patients with advance fibrosis. Patients with SVR have lower incidence of liver cirrhosis and HCC [3,4], and regression of liver stiffness has also been observed via non-invasive assessment in previous reports [5,6,7]. Consequently, DAA treatment for HCV has become the first-choice treatment. However, in chronic hepatitis C patients with HBV co-infection, reactivation of HBV during DAA treatment has been reported, with incidence ranging from 19–30% in patients with HBsAg seropositivity and 0.8–2.4% in resolved or occult HBV [8,9,10,11,12]; about 9% of patients developed clinical hepatitis with overt elevation of ALT level [8], which causes liver inflammation and deteriorating liver fibrosis.

Whether the co-existence of HBV infection has an adverse impact on liver stiffness regression as well as other clinical outcomes in CHC patients co-infected with HBV in contrast to those with HCV mono-infection remains unclear. In this study, we aim to compare liver stiffness regression between patients with mono-HCV and HBV co-infection who achieved SVR after DAA treatment by adjustment of pretreatment liver stiffness status using propensity score matching method.

## 2. Materials and Methods

### 2.1. Subjects

A prospectively collected cohort of 2916 chronic hepatitis C (CHC) patients receiving DAAs were recruited during March 2015 to December 2019 in Chang Gung Memorial Hospital, Linkou branch, Taiwan. Patients with available HBsAg status before treatment and valid liver stiffness measurements (LSM) before and after DAA treatment were recruited for analysis. HBV co-infection was defined as HBsAg positive before treatment. LSM, assessed by transient elastography (TE, Fibroscan), was prospectively recorded at baseline and after SVR. The exclusion criteria were inability to complete DAA treatment course, no SVR, human immunodeficiency virus (HIV) co-infection, or age < 18 years old when starting DAA treatment. (Appendix A). Since the prophylactic use of nucleos(t)ide analogue (Nuc) among HBV/HCV co-infected patients receiving DAAs was not reimbursed by Taiwan health insurance bureau, the 52 HBV/HCV patients in this study were not prophylactically treated with Nucs. This study was approved by the IRB committee of Chang Gung Memorial Hospital, Linkou Medical Center, Taiwan (IRB Number: 201801572B0C101).

### 2.2. Data Acquisition

The pretreatment characteristics, including age, sex, body mass index (BMI), HCV genotype and viral load, alanine aminotransferase (ALT), total bilirubin, albumin, platelet count, international normalized ratio (INR), alpha-fetoprotein (AFP), and metabolic factors such as glycohemoglobin (HbA1c), total cholesterol, triglyceride, LDL, HDL, HOMA index and steatosis were prospectively recorded and retrospectively analyzed and compared between CHC patients co-infected with HBV and CHC mono-infected patients.

### 2.3. Follow-Up and Outcomes

Patients were monitored biweekly in the first month of DAA treatment followed by monthly monitoring of liver biochemistry until the 8–12 weeks of DAA therapy ended. SVR was assessed with HCV RNA detectability at 12 weeks after DAA end-of-treatment. Patients were followed up every 3–6 months thereafter. The primary outcome of this study was liver stiffness regression after SVR, assessed by transient elastography (TE, Fibroscan). The secondary outcomes were hepatic adverse events including hepatic decompensation and/or de-novo HCC development after SVR.

### 2.4. Definition

The definition of HBV co-infection is seropositive HBsAg before DAA treatment. HBV DNA was quantified by The Roche COBAS^®^ AmpliPrep/COBAS^®^ TaqMan^®^ HBV Test, v2.0. The definition of inactive HBV carrier is HBV DNA ≤2000 IU/mL before DAA treatment. HBV reactivation was defined as an HBV DNA increase of 2 log from baseline or rising above 100 IU/mL from undetectable status [11]. HBV-related hepatitis is defined as viral reactivation with abrupt elevation of serum ALT to >5× upper limit of normal. Advanced fibrosis is defined as fibroscan >9.5 kPa while cirrhosis is defined as fibroscan >12.5 kPa [13,14]. Pre-DAA liver biopsy was done in only 4 patients. The definition of hepatic decompensation includes jaundice with bilirubin >3.0 mg/dL, coagulopathy with INR >2.0, new diagnosed ascites by liver echo, new esophageal varices requiring endoscopic treatment, or hepatic encephalopathy diagnosed by clinical physicians. Hepatocellular carcinoma was diagnosed by liver dynamic contrast-enhanced computed tomography and/or liver biopsy.

### 2.5. Non-Invasive Liver Stiffness Assessment

Liver stiffness was measured by means of transient elastography (FibroScan; Echosens, Paris, France) with an M- or XL-probe performed by well-trained technicians on the right lobe of liver. Baseline fibroscan data were obtained within 1 year before the start of DAA treatment (median 2.14 (IQR: 0.93–5.14) weeks before treatment). Post DAA assessment was performed after end of treatment (EOT) with median interval of 24.0 (IQR: 24.0–25.0) weeks). Results were expressed in kilopascal (kPa). Valid results of transient elastography consist of 10 successful acquisitions, with a success rate above 60% and an interquartile range (IQR) lower than 30% of the median value. LSM less than or equal to 7.1 kPa was considered as fibrosis stage F0–F1 by Metavir score. LSM between 7.1–9.5 kPa was considered to be fibrosis stage F2. LSM between 9.6–12.5 kPa was considered to be fibrosis stage F3. LSM greater than 12.5 kPa was considered to be fibrosis stage F4 by Metavir score.

### 2.6. Statistics

Continuous variables are expressed as medians (range) or mean ± (standard deviation), and categorical variables are expressed as frequencies (percentages). Mann–Whitney U test and Chi-square test were used to compare baseline characteristics between mono-HCV and HCV/HBV co-infected groups before DAA therapy. Wilcoxon signed-rank test was used to evaluate the improvement in each sub-group before and after DAA treatment. Multivariable linear regression analysis was carried out to identify factors associated with LSM improvement. All tests were 2-sided, and a *p* value < 0.05 was considered statistically significant. Propensity score matching (PSM) was used to eliminate heterogeneity between mono-HCV and HBV/HCV co-infection groups including age, gender, HCV genotype, pre-DAA treatment Metavir class, pre-DAA treatment platelet count and ALT. All analyses were performed using SAS software version 9.4 (SAS Institute, Inc., Cary, NC, USA).

## 3. Results

### 3.1. Overall Study Subjects

A total of 906 CHC patients with SVR were included in the current analysis. The overall characteristics are listed in Appendix A. The overall mean age was 63.2 (±11.7) years old, of which 39.7% were male. Co-infection with HBV was observed in 52 patients (5.7%). Of 37 of the 52 co-infected patients whose HBeAg statuses were available before DAA regimen, all showed HBeAg negative status. This is compatible with the phenomenon that the chronic hepatitis B prevalence rate in young people has largely decreased after nationwide neonatal vaccination programs since 1986; a growing proportion in HBV e antigen (HBeAg)-negative population among CHB patients has been observed [15]. Patients with HBV/HCV co-infection had a greater proportion of males (53.8% vs. 38.9%, *p* = 0.032), relatively younger age (61.8 vs. 63.2, *p* =0.317), and lower proportion of HCV genotype 1 infection (59.6% in HBV/HCV co-infected patients, vs. 64% in mono-HCV infected patients, *p* = 0.518) compared with HCV mono-infection groups. Pre-DAA ALT level (*p* = 0.514) and FIB-4 level (*p* = 0.393) were comparable between two groups, while LSM was numerically lower in HBV/HCV patients comparing to the mono-HCV arm (HBV/HCV versus mono-HCV: 8.15 kPa vs. 10.2 kPa, *p* = 0.091). None of the 52 HBV/HCV patients were treated with Nucs prophylactically, due to Taiwan’s reimbursement policy.

### 3.2. Liver Stiffness Regression after DAA Treatment and Its Predictors

Overall, significant improvement of LSM was observed in patients after DAA treatment (LSM before versus after DAA, median 10.15 kPa versus 6.9 kPa, *p* < 0.001) (Appendix A). Post-treatment LSM (HBV/HCV vs. mono-HCV: 6.2 kPa vs. 6.9 kPa, *p* = 0.281) and FIB-4 level (HBV/HCV vs. mono-HCV: 2.1 vs. 2.48, *p* = 0.2887) were comparable between the two groups (Table 1) while ALT levels appeared numerically higher in HBV/HCV co-infected patients (median 22 U/L vs. 18 U/L, *p* = 0.03) than in mono-HCV infected patients. A numerically lower reduction in LSM changes after DAA treatment was also observed in the HBV/HCV co-infected arm than the mono-HCV infected arm (median −0.85 kPa vs. −1.9 kPa, *p* = 0.094), although this did not reach statistical significance.

Of the total cohort of 906 patients, multivariate linear regression analysis, baseline ALT level (beta coefficient: −0.006 (95% −0.011 to −0.001), *p* = 0.016), total bilirubin (beta coefficient: 1.019 (95% 0.220–1.962), *p* = 0.014), albumin (beta coefficient: −1.318 (95% −2.548 to −0.088), *p* = 0.046), AFP (beta coefficient: −0.004 (95% −0.008 to 0), *p* = 0.041), platelet count (beta coefficient: −0.013 (95% −0.021 to −0.006), *p* < 0.001) and baseline LSM (beta coefficient: −0.33 (95% −0.372 to −0.287), *p* < 0.001) were the independent factors for LSM improvement (Table 2).

### 3.3. Liver Stiffness Regression Comparison in the Matched Study Cohort

Since previous studies showed higher baseline LSM correlated with greater LSM improvement after treatment [16], it is important to adjust the baseline LSM in HCV mono-infected and HBV/HCV co-infected patients to avoid confounders. Propensity score matching (PSM) adjusting for pre-DAA therapy Metavir fibrosis score, age, gender, HCV genotype (1 or non-1), pre-DAA therapy ALT and platelet count at 1 to 3 ratio of HBV/HCV versus mono-HCV infected groups was done to minimize characteristic differences (Table 3). After PSM with 1 to 3 ratio, a total of 208 patients (52 patients in the HBV/HCV co-infection arm, 156 patients in the mono-HCV infection group) were enrolled in this matched analysis. Patients with HBV/HCV co-infection showed less reduction of LSM changes after DAA treatment (median −0.85 kPa vs. −1.65 kPa, *p* = 0.251) than mono-HCV infected patients, although this did not achieve statistical significance (Table 4 and Appendix A).

### 3.4. Impact of HBV Viremia on Liver Stiffness Regression in the Matched Cohort

To investigate the influence of HBV viremia status on liver stiffness regression, patients with HBV/HCV co-infection were further stratified into inactive/active carriers by pre-DAA HBV DNA level < or ≥2000 IU/mL (Appendix A). Among HBV/HCV co-infection group, the active carriers (n = 7) have numerically higher post treatment ALT (29.0 U/L vs. 21 U/L vs. 18 U/L, *p* = 0.08) and less LSM reduction (−0.6 kPa vs. −0.8 kPa vs. −1.7 kPa, *p* = 0.234) compared with mono-HCV and inactive HBV carriers (n = 42) (Figure 1).

### 3.5. Clinical Outcome during Follow-Up

Among patients with HBV/HCV co-infection who achieved SVR, 2/52 patients suffered HBV flare up and received Nucs therapy during DAA treatment at week 9 and 8, with maximum ALT levels of 1584 U/L and 945 U/L without hyperbilirubinemia, respectively. No hepatic decompensation was recorded in these two patients. Both of the patients had detectable HBV DNA before DAA treatment (HBV DNA level of 894 IU/mL, 58,406 IU/mL, and pre-DAA ALT level of 46 U/L, 14 U/L respectively). Both patients’ liver biochemistry soon recovered to normal after application of ETV. One of the patients had increased LSM after DAA treatment (7 to 7.9 kPa), while the other patient showed no change in LSM after treatment (5.8 to 5.8 kPa). The characteristics and clinical course of these two patients are shown in Appendix A and Figure 2.

Over a median of 24.5 months follow-up after DAA treatment, 3 of the 52 (5.77%) HBV/HCV patients developed hepatic decompensation at month 36, 6 and 42 due to advanced cirrhosis rather than HBV reactivation, since ALT level remained consistently normal and there was no evidence of acute exacerbation. All three patients had cirrhosis before DAA treatment (pre-DAA LSM level: 48, 17.6, 28.4 kPa) (Appendix A). The rate of hepatic decompensation in HBV/HCV co-infected patients was numerically higher than that in the mono-HCV infection group (2/156, 1.26%) (*p* = 0.0673). The incidence rate of de-novo HCC is comparable between HBV/HCV and mono-HCV infected groups (HBV/HCV vs. mono-HCV: 1/45 (2.2%) vs. 5/136 (3.67%), *p* = 0.6366).

## 4. Discussion

In the era of DAA treatment, most patients achieve SVR, suggesting strong suppression of HCV activity. Nevertheless, a recent meta-analysis [11] reported 24% HBV reactivation and 9% reactivation-related hepatitis during DAA therapy, and this disease activity influenced liver stiffness values as measured by transient elastography [17]. To the best of our knowledge, however, no previous study has used transient elastography to compare liver stiffness regression of patients with mono-HCV and patients with HBV/HCV after DAA treatment.

In our study, baseline LSM is the most important factor correlated with liver stiffness regression after treatment in all patients. We found that the higher the baseline LSM, the greater the improvement measured after treatment, a result consistent with previous research [16]. After matching baseline LSM and other associated factors, patients with HBV/HCV co-infection (n = 52) have similar liver stiffness improvement compared with the HCV mono-infection group (n = 156) (median −0.85 kPa vs. −1.65 kPa, *p* = 0.251).

Since only patients who achieved SVR after DAA treatment were recruited in this study, the difference in LSM improvement may result from HBV activities. Previous studies have demonstrated most HBV = related hepatitis events only occur among patients with detectable baseline HBV DNA [11]. Since there lacks a universal consensus on the prophylactic use of Nucs in HBV/HCV co-infected patients undergoing DAA therapy and the APASL guidelines also recommend preemptive rather than prophylactic use of Nuc [18,19], only two patients who encountered HBV reactivation with hepatitis flare received pre-emptive ETV treatment under Taiwan’s treatment policy. These two reactivated patients had detectable HBV DNA before DAA therapy. However, the LSM changes between pre-DAA therapy HBV DNA level <2000 vs. ≥2000 IU/mL were comparable (−0.6 kPa vs. −0.8 kPa, *p* = 0.466), implying that the impact of HBV reactivation during DAA treatment on liver stiffness may be trivial, especially when this reactivation consists of HBV viremia alone without accompanying biochemical breakthroughs such as hepatitis flare. Both HBV/HCV co-infected patients in this study receiving Nuc treatment on demand due to HBV flare up had satisfactory HBV DNA suppression and ALT recovery, leading to minimal impact on liver stiffness progression. No liver stiffness regression by LSM was observed in these two cases 24 weeks after EOT. The insignificant difference comparing the LSM changes and adverse outcomes between patients with HBV/HCV co-infection and mono-HCV infection provides the evidence that the issue of HBV/HCV co-infection has limited impact on patients’ liver adverse outcomes in the era of DAA treatment. Hepatic decompensation principally developed after DAA treatment due to cirrhotic status prior to therapy rather than due to HBV co-infection. Cirrhosis is still the most important factor for clinical outcome.

There were several limitations to our study. First, HBV DNA levels were not prospectively assayed due to the Taiwan healthcare reimbursement policy in this prospective cohort study with retrospective analysis. The lack of a prospectively HBV DNA assessment may underestimate the incidence of HBV reactivation, especially those with log increase in viral load alone without biochemical breakthrough. Thus, we could only evaluate the impact of HBV co-existence, rather than the incidence of HBV reactivation, on liver stiffness regression. Second, only patients who had completed DAA treatment with SVR were included in this analysis. As some patients discontinued treatment due to other reasons than HBV, such as sepsis or non-liver-related mortality, the actual incidence rate of HBV-related events may be underestimated. Third, since the prevalence of hepatitis D (HDV) was not checked in this study, the potential influence of HDV could not be evaluated. Fourth, since liver stiffness has been reported to regress earlier and faster than histological fibrosis [20], liver stiffness should not be taken to precisely indicate actual liver fibrosis. Fifth, the median follow-up of this study was only 2 years, and only a few patients developed adverse hepatic events during follow-up, making it difficult to assess long term clinical outcomes, especially HCC incidence, for each group. Sixth, since none of the cases involved prophylactic use of Nucs, the efficacy of Nucs usage in ensuring liver stiffness regression could not be assessed by the current study. Seventh, since this study recruited HBV/HCV co-infected patients with HBsAg seropositivity, the issue of HBV reactivation in resolved or occult HBV infection is beyond the scope of current study. Finally, the number of HBV-HCV co-infection cases is not sufficient, and our findings need to be supported by evidence from a larger cohort before firmly concluding HBV co-infection has minimal impact on CHC patient outcomes.

## 5. Conclusions

Our study suggests that liver stiffness regression occurs in both mono-HCV and HBV/HCV co-infection patients receiving DAA treatment. Co-infection with HBV has limited influence on liver stiffness regression after DAA treatment. Pre-DAA cirrhotic status remains the most important factor in the occurrence of hepatic adverse events after SVR. The short-term impact (<3 years) of HBV/HCV co-infection on liver stiffness changes and liver adverse events after DAA treatment is negligible while the long-term effect (>3 years) remains unknown.

## Figures and Tables

**Figure 1 viruses-14-00786-f001:**
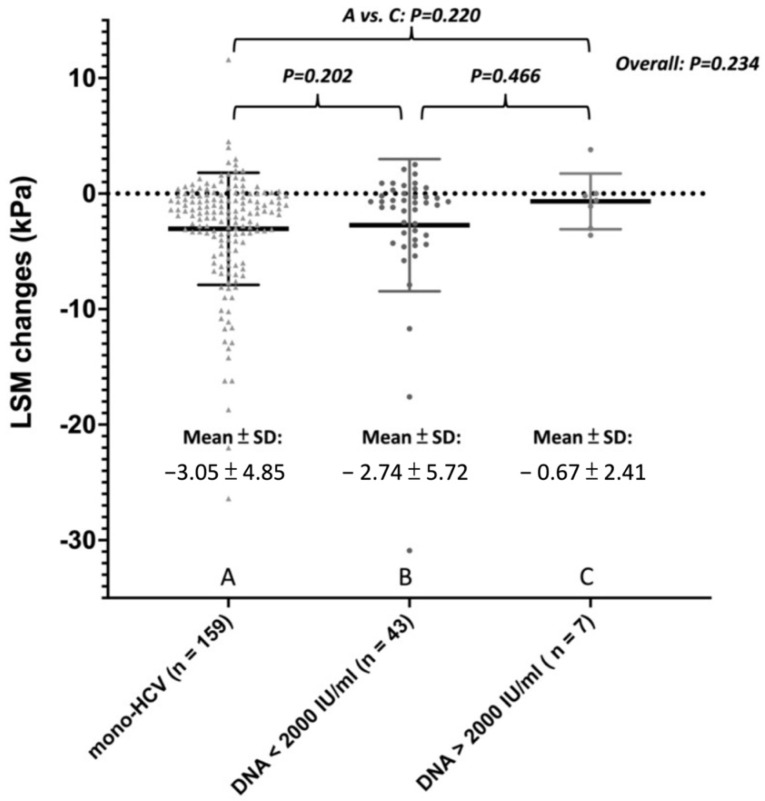
Liver stiffness measurements change after SVR among patients with mono-HCV, co-infection with inactive HBV, or co-infection with active HBV.

**Figure 2 viruses-14-00786-f002:**
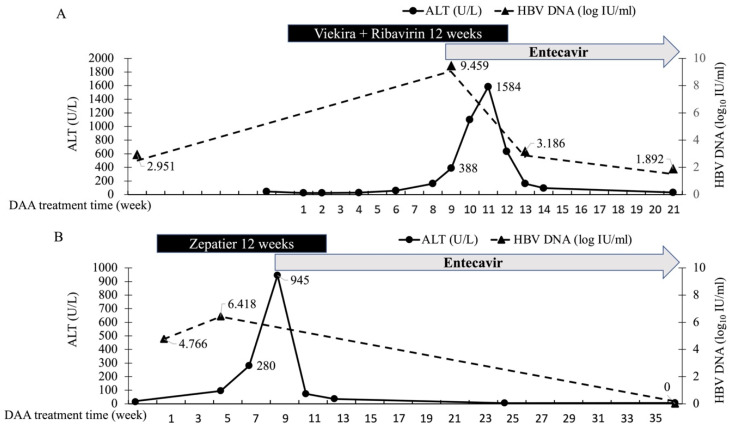
ALT and HBV DNA level in patients develops HBV flare during DAA Treatment. (**A**) A 52 years old male, non-cirrhotic, encountered asymptomatic ALT elevation up to 1584U/L at week 8 of Viekira+Ribavirin therapy. Hepatitis event timely improved after Entecavir application. (**B**) A 48 years old male, no cirrhosis, encountered asymptomatic ALT surge up to 945 U/L at week 6 of Zepatir treatment. Hepatitis event timely improved after Entecavir treatment at week 9.

**Table 1 viruses-14-00786-t001:** Post-treatment demographic comparison between patients with dual infection and mono-HCV.

		Mono-HCV or B + C Co-Infection		
Variables	All (N = 906)	Mono-C (N = 854, 94%)	B + C (N = 52, 6.0%)	*p* Value
Post DAA FIB-4 score	2.475 (0.41~60.06)	2.48 (0.48~60.06)	2.11 (0.41~6.28)	0.289
FIB4 < 1.45	142 (16.32%)	132 (16.08%)	10 (20.41%)	0.667
1.45 ≤ FIB4 < 3.25	453 (52.07%)	430 (52.38%)	23 (46.94%)	
3.25 ≤ FIB4	275 (31.61%)	259 (31.55%)	16 (32.65%)	
FIB4_change	−0.265 (−21.0~53.3)	−0.26 (−21.0~53.3)	−0.28 (−9.74~1.39)	0.856
Post DAA LSM	6.9 (2.6~75.0)	6.9 (2.6~75.0)	6.2 (3.3~46.4)	0.282
LSM < 7.1	470 (51.88%)	441 (51.64%)	29 (55.77%)	0.431
7.1 ≤ LSM < 9.5	139 (15.34%)	133 (15.57%)	6 (11.54%)	
9.5 ≤ LSM < 12.5	93 (10.27%)	85 (9.95%)	8 (15.38%)	
12.5 ≤ LSM	204 (22.52%)	195 (22.83%)	9 (17.31%)	
LSM decrease	−1.8 (−49.5~32.6)	−1.9 (−49.5~32.6)	−0.85 (−30.9~3.8)	0.094
LSM improve > 30%	331 (36.53%)	319 (37.35%)	12 (23.08%)	0.038
LSM improve ≤ 30%	575 (63.47%)	535 (62.65%)	40 (76.92%)	

ALT, alanine aminotransferase; FIB4, the Fibrosis-4 index; HBV, hepatitis B virus; HCV, hepatitis C virus; LSM, liver stiffness measurement.

**Table 2 viruses-14-00786-t002:** Linear regression analysis of factors associated with LSM change.

	Univariate					Multivariate				
Variables	Effect	95% CI		SE	*p*-Value		95% CI		SE	*p*-Value
Age (per year)	−0.027	−0.061	0.007	0.017	0.122					
Male gender	−0.441	−1.261	0.379	0.418	0.292					
Genotype non-1	0.794	−0.041	1.628	0.425	0.062					
HCV RNA (Log_10_IU/mL)	0.042	−0.400	0.485	0.225	0.851					
BMI	−0.022	−0.128	0.083	0.054	0.678					
HbA1c	−0.334	−0.726	0.057	0.199	0.094					
HOMA	−0.083	−0.184	0.018	0.051	0.107					
Cholesterol	0.025	0.011	0.039	0.007	<0.001	0.022	−0.005	0.050	0.014	0.111
Triglyceride	0.002	−0.007	0.012	0.005	0.617					
LDL (mg/dL)	0.023	0.008	0.039	0.008	0.003	−0.006	−0.038	0.026	0.016	0.720
ALT (U/L)	−0.016	−0.021	−0.011	0.003	<0.001	−0.006	−0.011	−0.001	0.003	0.016
T-bil. (mg/dL)	−1.015	−1.863	−0.166	0.432	0.019	1.091	0.220	1.962	0.444	0.014
Albumin (g/dL)	2.956	1.947	3.965	0.514	<0.001	−1.318	−2.548	−0.088	0.627	0.036
INR	−8.240	−11.654	−4.826	1.739	<0.001	−0.179	−3.897	3.540	1.894	0.925
AFP (ng/mL)	−0.009	−0.013	−0.005	0.002	<0.001	−0.004	−0.008	0.000	0.002	0.041
Platelet (10^3^/uL)	0.015	0.009	0.021	0.003	<0.001	−0.013	−0.021	−0.006	0.004	<0.001
FIB-4 score	−0.304	−0.405	−0.202	0.052	<0.001					
Baseline LSM	−0.283	−0.313	−0.253	0.015	<0.001	−0.330	−0.372	−0.287	0.022	<0.001
HBV co-infection	0.480	−1.246	2.207	0.879	0.585					

AFP, alpha-fetoprotein; ALT, alanine aminotransferase; AST, aspartate aminotransferase; BMI, body mass index; 95%CI, 95% confidence interval; DAA, direct antiviral agents; FIB4, The Fibrosis-4 Index; HbA1c, glycohemoglobin; HBV, hepatitis B virus; HCV, hepatitis C virus; HOMA, Homeostatic Model Assessment for Insulin Resistance; INR, international normalized ratio; LDL: low-density lipoprotein; LSM: liver stiffness measurement; SE, standard error; T-bil.: total bilirubin; ULN, ULN, upper limit of normal (ULN of ALT = 36 U/L).

**Table 3 viruses-14-00786-t003:** Pretherapy demographic comparison after PSM (1:3 match, n = 208).

		Mono HCV or B + C Co-Infection		
Variables	All (N = 208)	Mono-C (N = 156)	B + C (N = 52) *	*p* Value
Age (years)	62.8 (26.4~91.9)	62.85 (31.6~91.9)	61.85 (26.4~84.4)	0.413
Male, n (%)	112 (53.85%)	84 (53.85%)	28 (53.85%)	1.0
Cirrhosis by echo, n (%)	54 (25.96%)	40 (25.64%)	14 (26.92%)	0.855
Genotype, n (%)				
Non-1	67 (32.21%)	46 (29.49%)	21 (40.39%)	0.145
1	141 (67.79%)	110 (70.51%)	31 (59.62%)	
HCV RNA (Log_10_IU/mL)	6.22 (3.23~7.29)	6.21 (3.23~7.29)	6.22 (3.7~7.04)	0.344
≥6, n (%)	119 (57.49%)	89 (57.05%)	30 (58.82%)	0.824
<6	88 (42.51%)	67 (42.95%)	21 (41.18%)	
BMI (kg/m^2^)	24.4 (16.8~37.5)	24.5 (17.07~37.5)	24.4 (16.8~32.89)	0.392
HbA1c	5.7 (4.6~11.9)	5.7 (4.6~11.9)	5.75 (4.9~11.3)	0.775
≥6.5, n (%)	37 (18.69%)	26 (17.33%)	11 (22.92%)	0.388
<6.6	161 (81.31%)	124 (82.67%)	37 (77.08%)	
HOMA index	2.07 (0.38~58.29)	2.16 (0.48~19.71)	1.97 (0.38~58.29)	0.354
≥2.5, n (%)	51 (35.17%)	40 (37.38%)	11 (28.95%)	0.35
<2.5	94 (64.83%)	67 (62.62%)	27 (71.05%)	
Cholesterol (mg/dL)	170.0 (97.0~283.0)	166.0 (115.0~283.0)	176.0 (97.0~230.0)	0.54
≥200, n (%)	33 (20.12%)	26 (20.80%)	7 (17.95%)	0.698
<200	131 (79.88%)	99 (79.20%)	32 (82.05%)	
Triglyceride (mg/dL)	92.0 (34.0~299.0)	94.0 (34.0~275.0)	91.0 (39.0~299.0)	0.524
≥100, n (%)	69 (42.07%)	54 (43.20%)	15 (38.46%)	0.601
<100	95 (57.93%)	71 (56.80%)	24 (61.54%)	
LDL (mg/dL)	103.0 (36~191)	102.0 (47~191)	109 (36~152)	0.703
≥130, n (%)	34 (21.38%)	27 (22.13%)	7 (18.92%)	0.676
130	125 (78.62%)	95 (77.87%)	30 (81.08%)	
ALT (U/L)	49.0 (8.0~324.0)	47.0 (8.0~273.0)	56.0 (12.0~324.0)	0.592
<1 × ULN, n (%)	67 (32.21%)	53 (33.97%)	14 (26.92%)	0.484
1–5×	130 (62.50%)	96 (61.54%)	34 (65.39%)	
≥5×	11 (5.29%)	7 (4.49%)	4 (7.69%)	
T-bil. (mg/dL)	0.7 (0.2~3.4)	0.7 (0.2~3.4)	0.8 (0.2~1.7)	0.124
Albumin (g/dL)	4.31 (2.88~5.15)	4.28 (3.39~5.15)	4.325 (2.88~4.8)	0.682
INR ≦ 1.3, n (%)	204 (99.03%)	153 (99.35%)	51 (98.08%)	0.418
>1.3	2 (0.97%)	1 (0.65%)	1 (1.92%)	
AFP (ng/mL)	3.55 (1.0~248.1)	3.65 (1.0~248.1)	3.45 (2.0~205.0)	0.652
Platelet (10^3^/uL)	178 (29~384)	178.0 (29~384)	179.5 (42~338)	0.18
FIB-4 score	2.34 (0.29~20.18)	2.34 (0.29~20.18)	2.37 (0.41~16.38)	0.416
FIB4 < 1.45, n (%)	42 (20.19%)	31 (19.87%)	11 (21.15%)	0.396
1.45 ≤ FIB4 < 3.25	96 (46.15%)	76 (48.72%)	20 (38.46%)	
3.25 ≤ FIB4	70 (33.65%)	49 (31.41%)	21 (40.39%)	
LSM	7.7 (3.4~52.3)	7.7 (3.4~52.3)	8.15 (3.6~48.0)	0.993
LSM < 7.1, n (%)	92 (44.23%)	69 (44.23%)	23 (44.23%)	1.000
7.1 ≤ LSM < 9.5	36 (17.31%)	27 (17.31%)	9 (17.31%)	
9.5 ≤ LSM < 12.5	28 (13.46%)	21 (13.46%)	7 (13.46%)	
12.5 ≤ LSM	52 (25.0%)	39 (25.0%)	13 (25.0%)	
End of treatment ALT	19.0 (6.0~635.0)	18.0 (6.0~173.0)	22.0 (6.0~635.0)	0.0502

The continuous variables are expressed as medians (range) and categorical variables are expressed as frequencies (percentages). AFP, alpha-fetoprotein; ALT, alanine aminotransferase; AST, aspartate aminotransferase; BMI, body mass index; DAA, direct antiviral agents; HbA1c, glycohemoglobin; HBV, hepatitis B virus; HCV, hepatitis C virus; HOMA, Homeostatic Model Assessment for Insulin Resistance; IFN, interferon; INR, international normalized ratio; T-bil.: total bilirubin; ULN, upper limit of normal (ULN of ALT: 36 U/L); ×, times. * HBeAg status is available in 37/52 patients in co-infection group; all the 37 patients were HBeAg negative.

**Table 4 viruses-14-00786-t004:** Post-treatment demographic comparison after PSM. Stratified by pre-DAA LSM status.

		Mono HCV or B + C Co-infection		
Variables	No.	Mono-C	B + C	*p* Value
Overall	N = 208	N = 156	N = 52	
LSM change	−1.5 (−30.9~11.6)	−1.65 (−26.4~11.6)	−0.85 (−30.9~3.8)	0.251
LSM improve > 30%	61 (29.33%)	49 (31.41%)	12 (23.08%)	0.253
LSM improve ≤ 30%	147 (70.67%)	107 (68.59%)	40 (76.92%)	
F0–2	N = 128	N = 96	N = 32	
LSM change	−0.7 (−4.6~4.5)	−0.7 (−4.1~4.5)	−0.6 (−4.6~2.5)	0.370
LSM improve > 30%	23 (17.97%)	19 (19.79%)	4 (12.50%)	0.352
LSM improve ≤ 30%	105 (82.03%)	77 (80.21%)	28 (87.50%)	
F3–4	N = 80	N = 60	N = 20	
LSM change	−5.3 (−30.9~11.6)	−6.0 (−26.4~11.6)	−3.8 (−30.9~3.8)	0.246
LSM improve > 30%	38 (47.50%)	30 (50.0%)	8 (40.0%)	0.438
LSM improve ≤ 30%	42 (52.50%)	30 (50.0%)	12 (60.0%)	

ALT, alanine aminotransferase; HBV, hepatitis B virus; HCV, hepatitis C virus; LSM, liver stiffness measurement.

## Data Availability

Data available on request due to restrictions eg privacy or ethical. The data presented in this study are available on request from the corresponding author. The data are not publicly available without the permission of IRB committee of Chang Gung Memorial Hospital, Linkou Medical Center.

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
