# Peer review of "Hepatitis B Co-Infection Has Limited Impact on Liver Stiffness Regression in Chronic Hepatitis C Patients Treated with Direct-Acting Antivirals"

_viruses, 2022, doi:10.3390/v14040786_

Round 1

Reviewer 1 Report

General comment

The problem of HBV reactivation under DAA therapy of chronic hepatitis C is well known and was already subject of several studies including one very recent from Taiwan (ref. 10). However, this study is inasmuch new as it puts its focus on the improvement of liver fibrosis under antiviral HCV therapy with or without HBV co-infection. The study design is adequate and most of the data are well presented. Some points should be corrected (1 – 6) or considered (7, 8).

Specific points

  1. The text defined HBV reactivation. But the results do not show data on reactivation. If there was no reactivation without hepatitis, this should be explicitly mentioned in para. 3.5.
  2. For HCV genotypes are distinguished by Arabic numerals: genotype 1, not genotype I.
  3. L141 and legend to table 1. How many months after DAA was the LSM measured? Directly or later?
  4. Table 3.
    1. The legend should explain what the numbers before and within the brackets mean.
    2. The value of the ULN for ALT should be given in the footnote next to the explanation of the ULN.
    3. The meaning of P-value of 0.035 for INR is not clear. There is obviously no difference between the two groups
  5. 1 is unsatisfactory. The meaning of the boxes (25 and 75 % percentils?) and the whiskers (3 s.d?) is not explained. The whiskers do not make sense. An LSM increase of >4 after therapy did obviously not occur and a decrease of -10 is unlikely if <7 is considered F0. It would be more informative to show scatter plots for each patient plus the median and 1x s.d.
  6. The Nuc treatment was not pre-emptive because it was given after the rise of the ALT.
  7. The discussion may briefly comment on the fact that all coinfected patients with HBeAg results available were negative. Is this normal for monoinfected patients in Taiwan?
  8. It should also mention that HBV reactivation may also occur in occult or resolved HBV infection.

Author Response

Response to Reviewer 1 Comments:

  1. The text defined HBV reactivation. But the results do not show data on reactivation. If there was no reactivation without hepatitis, this should be explicitly mentioned in para. 3.5

Response 1. Thank you for your kind comment. As a retrospective study, HBV DNA level were not prospectively and regularly assayed in this study due to the restriction of Taiwan's healthcare reimbursement policies which HBV DNA assessment be reimbursed only at timepoint of ALT elevation. Thus, the limitation of current study is that we could only report the incidence of clinically significant HBV reactivation which accompanied with ALT elevation but the incidence of HBV DNA reactivation alone with viral load > 1 log increase will not be able to report for the possible underestimation.However, virological reactivation is a very important issue. In regards to this limitation, we addressed this point in the limitation paragraph in the discussion section / L263.

  1. For HCV genotypes are distinguished by Arabic numerals: genotype 1, not genotype I.

Response 2. We will correct genotype I to genotype 1 in L131

  1. L141 and legend to table 1. How many months after DAA was the LSM measured? Directly or later?

Response 3. The LSM measurement was performed about 24 weeks after end-of-treatment (EOT), but not at the EOT. We will address this point in the method section 2.5: Non-invasive liver stiffness assessment, L103 and the instructions of Table 1.

  1. (1). Table 3. The legend should explain what the numbers before and within the brackets mean.

Response 4-1. Thanks for your kind reminding. We add the explain of numbers in the table to avoid obfuscate. The continuous variables are expressed as medians (range), and categorical variables are expressed as frequencies (percentages)

    4. (2). The value of the ULN for ALT should be given in the footnote next to the explanation of the ULN.

Response 4-2: The upper limit of normal of ALT in Chang Gung Memorial Hospital is 36U/N. We added the value of ULN in the tables.

    4. (3). The meaning of P-value of 0.035 for INR is not clear. There is obviously no difference between the two groups.

Response 4-3. Since most of the patient has clinical insignificant INR level (0.9~1.3), we divided INR level to normal group (INR =<1.3) ,  and PT prolonged group (1.3 < INR) according to clinical relevance instead of using continuous variable, which would help us to avoid confusion. The distribution of PT prolongation is comparable between mono-HCV infection and HBV-HCV co-infection groups (1/154 in mono C and 1/52 in B+C coinfection, p = 0.418) We uodated the data in Table 3.

  1. 1 is unsatisfactory. The meaning of the boxes (25 and 75 % percentils?) and the whiskers (3 s.d?) is not explained. The whiskers do not make sense. An LSM increase of >4 after therapy did obviously not occur and a decrease of -10 is unlikely if <7 is considered F0. It would be more informative to show scatter plots for each patient plus the median and 1x s.d.

To make the figure more informative, we redraw the figure 1 with mean +/- standard deviation, outer points were present as scatter plots to make it.

  1. The Nuc treatment was not pre-emptive because it was given after the rise of the ALT

We will correct pre-emptive to on-demand Nuc treatment in the manuscript discussion section / L252

  1. The discussion may briefly comment on the fact that all coinfected patients with HBeAg results available were negative. Is this normal for monoinfected patients in Taiwan?

Since the neonatal HBV vaccination program starting from 1986, the chronic hepatitis B prevalence rate in young people has largely decreased and a growing proportion in HBV e antigen (HBeAg) negative population among CHB patients has been observed[1]. We addressed this in the manuscript result section L131

  1. It should also mention that HBV reactivation may also occur in occult or resolved HBV infection

HBV reactivation in occlut or resolved HBV infected patients during DAA treatment could happen, but the incidence was rare [1·4% (0·8-2·4)%] by recent meta-analysis (Mucke MM et al Lancet Gastroenterol Hepatol 2018; Mar;3(3):172-180].  Our study only enrolled the patient with HBsAg seropositivity whom supposed to have higher risk of HBV reactivation. The issue of reactivation in resolved/occlut HBV infected patients is beyond our study population which is addressed in the paragraph of limitation in discussion section /L279 and section of introduction/ L47

Reviewer 2 Report

Dear Authors,

The manuscript is well-written and has merit, however, the results obtained, for the most part, are not statistically relevant. Despite this, the proposal addressed is relevant and can serve as a good background for future studies.

Results:

-In tables 3 and 4, the total number of patients before PSM =208. The explanation for this reduction in the total number of evaluated individuals must be clarified in the manuscript.

Table S5 needs to be formatted in a more readable and organized way.

Results/Discussion:

-Most of the results have no statistical relevance. This must be discussed.

Author Response

Response to Reviewer 2’s comments

  1. In tables 3 and 4, the total number of patients before PSM =208. The explanation for this reduction in the total number of evaluated individuals must be clarified in the manuscript.

Response 1. Thank you for your reminding. We addressed the number of patient after PSM = 208 in the manuscript, result section L176

  1. Table S5 needs to be formatted in a more readable and organized way

Response 2. Thank you for your suggestion. We have formated Supplementary table 5 to make it more easily to understand the clinical course of patient with flare events and decompensation.

  1. Most of the results have no statistical relevance This must be discussed

Response 3. Although most of the comparison between patients with mono-HCV and HBV/HCV coinfection did not reach statistical significance, these findings may overthrow the original impression that HBV/HCV coinfection lead to worse outcome than HCV mono-infection patients in the era of DAA treatment. We would add the statement in discussion section L257

Round 2

Reviewer 2 Report

Dear authors.

The manuscript has improved considerably, so I consider the paper suitable for publication.